# Evaluation of the Durability of Slippery, Liquid-Infused Porous Surfaces in Different Aggressive Environments: Influence of the Chemical-Physical Properties of Lubricants

Federico Veronesi *, Guia Guarini, Alessandro Corozzi and Mariarosa Raimondo

National Research Council—Institute of Science and Technology for Ceramics CNR-ISTEC, 48022 Faenza, Italy; istec@istec.cnr.it (G.G.); alessandro.corozzi@istec.cnr.it (A.C.); mariarosa.raimondo@istec.cnr.it (M.R.)
* Correspondence: federico.veronesi@istec.cnr.it; Tel.: +39-054-669-9727

**Abstract:** Liquid-repellent surfaces have been extensively investigated due to their potential application in several fields. Superhydrophobic surfaces achieve outstanding water repellence, however their limited durability in severe operational conditions hinders their large-scale application. The Slippery, Liquid-Infused Porous Surface (SLIPS) approach solves many of the durability problems shown by superhydrophobic surfaces due to the presence of an infused liquid layer. Moreover, SLIPS show enhanced repellence towards low surface tension liquids that superhydrophobic surfaces cannot repel. In this perspective, SLIPS assume significant potential for application in harsh environments; however, a systematic evaluation of their durability in different conditions is still lacking in the literature. In this work, we report the fabrication of SLIPS based on a ceramic porous layer infused with different lubricants, namely perfluoropolyethers with variable viscosity and n-hexadecane; we investigate the durability of these surfaces by monitoring the evolution of their wetting behavior after exposure to severe environmental conditions like UV irradiation, chemically aggressive solutions (acidic, alkaline, and saline), and abrasion. Chemical composition and viscosity of the infused liquids prove decisive in determining SLIPS durability; especially highly viscous infused liquids deliver enhanced resistance to abrasion stress and chemical attack, making them candidates for applicable, long-lasting liquid-repellent surfaces.

**Keywords:** liquid-infused surfaces; durability; lubricants; wetting; liquid-repellent coatings

## 1. Introduction

Liquid-repellent surfaces have drawn huge interest in the last years due to their inherent self-cleaning properties [1–3] and other potential related advantages e.g., anti- or de-icing properties [4–6], drag and friction reduction [7–9], and anti-fouling behavior [10,11]. All these properties might have positive fallout on a wide range of industrial fields and applications like aircraft, building materials, ships, machinery, and countless others. Therefore, the fabrication of liquid-repellent surfaces has become one of the hottest topics in the material science community. The first and most well-known approach to the fabrication of liquid-repellent surfaces starts from the mimicry of the dual-scale (i.e., hierarchical) surface structure observed on the lotus leaf [12]. By coupling the creation of micro/nanoscale surface features with the tailoring of surface chemical composition to achieve low surface energy, superhydrophobic surfaces can be obtained [13,14]. These materials are characterized by high water contact angles, i.e., larger than 150°, and high mobility of water drops that are placed on their surface. Such extreme water repellence is due to the trapping of air pockets between surface morphological features, leading to minimal adhesion between water drops and the solid surface. However, the lotus leaf-like surfaces show some limitations: for instance, they are not able to repel liquids with a surface tension lower than that of water (72 mN·m$^{-1}$ at 25 °C) like oils and alkanes. For that reason, the scientific community started to design surfaces with simultaneous repellence to

water (i.e., hydrophobicity) and other liquids, mainly non-polar oils (i.e., oleophobicity). Such combination is commonly termed as amphiphobicity [15,16] or, in case the repellence is extended also to complex fluids like blood or milk, omniphobicity [17,18].

Another limitation of the lotus-mimicking approach is that the trapped air pockets can be displaced by fast impinging drops or when the liquid pressure is too high. Perhaps the most promising fabrication approach to overcome this limit has been proposed by Wong et al. [19] in 2011 and takes inspiration from the surface of Nepenthes plant [20], which is covered by a layer of liquid lubricant that makes it extremely slippery for the insects that land on it. The proposed Slippery, Liquid-Infused Porous Surfaces (SLIPS) display much lower contact angles compared to superhydrophobic surfaces but drops of many liquids can still move very easily on them. Moreover, these materials possess much improved stability when submerged, along with other enhanced properties [21,22].

Due to such exciting potential, SLIPS have drawn large interest and many papers have focused on the design criteria to achieve the best omniphobicity [23]. In principle, SLIPS can be fabricated by using any liquid as infused lubricant, but three fundamental criteria for SLIPS design always hold: the fluid to be repelled must be immiscible with the infused lubricant; the lubricant must wet the solid structures completely, even when no outer liquid is present; the outer liquid must form discrete droplets on the SLIPS. Notwithstanding these design guidelines for SLIPS, the literature still lacks an assessment of their durability in different environments that simulate their performance in real operational conditions. Moreover, a systematic investigation of the relationship between infused lubricant and durability is missing.

In this paper, we report the fabrication of SLIPS infused with different lubricants and study the evolution of their wetting properties after testing in several conditions, namely UV irradiation, immersion in chemically aggressive solutions, and abrasion. Both fluorinated and fluorine-free lubricants were tested, as the former are known for inducing excellent oleophobicity but are also deemed to be environmentally harmful [24], while the latter can only generate hydrophobicity but are much more eco-friendly. The results provide relevant insight on the behavior of SLIPS and draw guidelines for the choice of the lubricants to be infused for long-lasting repellence.

## 2. Materials and Methods

### 2.1. SLIPS Fabrication

The nanostructured pseudoboehmite AlOOH coating was obtained by deposition of an alumina suspension synthesized via a previously reported sol-gel route [25]. Aluminum samples (Al 1050 alloy) were used as substrates; prior to deposition, they were cleaned with an ultrasonic bath in ethanol for 5 min and dried in air. The synthesized alumina nanoparticles were deposited on the substrates via dip-coating in controlled conditions, followed by treatment with boiling water. The complete procedure for the coating fabrication is reported elsewhere [26]. Pseudoboehmite-coated samples were observed by Scanning Electron Microscopy (FESEM Gemini Columns SIGMA, Carl Zeiss Microscopy GmbH, Oberkochen, Germany).

In order to increase the affinity between the nanostructured, porous coating and the infused lubricant, the former was chemically modified depending on the nature of the latter. When n-hexadecane was used as lubricant, alkyl chains were grafted to pseudoboehmite by immersion in a hexadecyltrimethoxysilane ($\geq$85%, Merck, Darmstadt, Germany) solution in ethanol (6 wt.%) for 2 min, followed by drying at 80 °C for 1 h and annealing at 170 °C for 5 min. Otherwise, when Perfluoropolyethers (PFPE) were used as lubricants, fluorinated alkyl chains were grafted to the nanostructured AlOOH by immersion in a commercial solution of fluoroalkylsilane in isopropyl alcohol (Protectosil SC200, Evonik, Essen, Germany) for 2 min, followed by drying in air and annealing at 150 °C for 30 min.

Finally, lubricants were infused in the nanostructured layer by brushing until a continuous, shiny layer covered the sample surface. In order to remove excess oil, the samples were held vertical overnight. Five different lubricants were used: four commercial perfluo-

ropolyethers PFPE with increasing viscosity (Krytox GPL 100, 103, 105, and 107, Chemours, Geneva, Switzerland), or n-hexadecane (≥99%, Merck). The resulting SLIPS were labeled as K100, K103, K105, K107, and HEX, respectively.

*2.2. SLIPS Characterization*

The wetting behavior of SLIPS was determined by measuring the Advancing (ACA) and Receding Contact Angles (RCA) with water and n-hexadecane drops to calculate Contact Angle Hysteresis (CAH) as the difference between the two. Advancing and receding contact angles and contact angle hysteresis with water and n-hexadecane are labeled as $ACA_W$, $RCA_W$, $CAH_W$, $ACA_H$, $RCA_H$, and $CAH_H$ in the text. We chose not to measure "static" contact angles, which are commonly reported in most papers about liquid-repellent surfaces, as they do not adequately describe the behavior of liquid drops on SLIPS. Indeed, static contact angles rarely exceed 120° on SLIPS, while the low CAH well shows the little adhesion of drops on the surface. Contact angle measurements were performed with an optical contact angle system (DSA 30S, Krüss GmbH, Hamburg, Germany). For all contact angle measurements, 5 µL drops were first dispensed with a software-controlled syringe and gently deposited on the surface. Then, 5 µL were added to the drop and the ACA measured during drop expansion. Finally, 5 µL were removed from the drop and the RCA measured to calculate CAH as the difference between ACA and RCA. For each surface, at least five different points were characterized to calculate average ACA, RCA, and CAH with related standard deviations.

The evolution of chemical composition of coated surfaces was monitored via Fourier Transform Infrared spectroscopy (FTIR) using a Nicolet IS5 spectrophotometer (ThermoFisher Scientific, Waltham, MA, USA) in the Attenuated Total Reflection mode. A diamond crystal was pressed against the samples; each spectrum was collected in the 4000–550 $cm^{-1}$ range with a resolution of 4 $cm^{-1}$ and averaged over 16 scans. For all spectra, the non-infused face of the pristine sample was used as background to obtain the spectrum of the infused liquid phase.

*2.3. Durability Tests*

SLIPS were subjected to three types of tests to assess their durability in different aggressive environments.

UV irradiation tests were performed exposing the samples to a lamp (UVGL-58, UVP International, Jena, Germany) with a radiation intensity of 2.0 mW $cm^{-2}$ measured at λ = 354 nm. The samples were collected, and their wetting properties assessed after 2, 4, 6, and 8 h of irradiation.

Chemical ageing tests were performed by immersing SLIPS in three different aqueous solutions: acidic (pH = 3, obtained by dilution of 3.3 g·$L^{-1}$ of acetic acid, ≥99%, Merck), alkaline (pH = 11, obtained by dissolution of 0.04 g·$L^{-1}$ of sodium hydroxide, 99%, Merck), and saline (obtained by dissolution of 100 g·$L^{-1}$ of sodium chloride ≥99%, Merck). After fixed amounts of time, samples were withdrawn from the solutions, rinsed with deionized water, dried in air, and characterized in terms of wetting properties, then re-immersed in the solutions.

Abrasion tests were performed as per UNI EN 1096-2 standard, which is used to assess the mechanical properties of coated glasses in the building industry. Samples were fixed to a technical balance with double-sided adhesive tape and a rotating felt disk (diameter 60 mm, rotational speed 60 rpm) was pushed against the sample until a weight of about 400 g, equivalent to a force of about 4 N, was measured by the balance. The abrasion continued for 30 s, then the sample was removed, and its wetting properties re-evaluated.

## 3. Results and Discussion

*3.1. SLIPS Characterization*

For all investigated SLIPS, the same nanostructured pseudoboehmite AlOOH coating was used to retain the infused oil [27]. Such coating is made of randomly oriented lamellae

(about 200 nm long and few nm thick) separated by 30–50 nm voids which serve as pockets for lubricant retention. The morphology of the, as-prepared, pseudoboehmite coating on aluminum is reported in Figure 1.

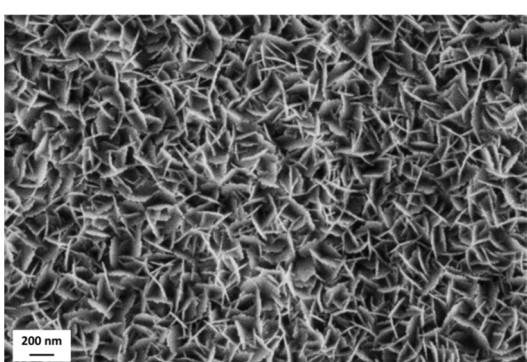

**Figure 1.** Scanning electron micrograph of the nanostructured pseudoboehmite layer. Scale bar is reported.

AlOOH coatings were infused with either PFPE or n-hexadecane. Table 1 reports all measured contact angle values for as-fabricated SLIPS.

**Table 1.** Kinematic viscosity of the infused oils at $T = 40\ °C$ and wetting data of related infused surfaces (ACA: Advancing Contact Angle, RCA: Receding Contact Angle, and CAH: Contact Angle Hysteresis; subscript W: Water and subscript H: Hexadecane). Standard deviations are reported as errors. Viscosity values were obtained from the suppliers.

| Infused Liquid | Viscosity [mm²/s] | $ACA_W$ [°] | $RCA_W$ [°] | $CAH_W$ [°] | $ACA_H$ [°] | $RCA_H$ [°] | $CAH_H$ [°] |
|---|---|---|---|---|---|---|---|
| Krytox 100 | 7.8 | $124.2 \pm 0.7$ | $122.2 \pm 0.4$ | $1.9 \pm 0.9$ | $71.4 \pm 0.4$ | $69.5 \pm 0.4$ | $1.9 \pm 0.5$ |
| Krytox 103 | 30 | $121.8 \pm 0.6$ | $117.7 \pm 1.3$ | $4.2 \pm 1.9$ | $70.7 \pm 0.4$ | $68.1 \pm 0.9$ | $2.6 \pm 0.6$ |
| Krytox 105 | 160 | $121.5 \pm 1.5$ | $115.6 \pm 0.9$ | $5.9 \pm 0.7$ | $71.8 \pm 1.1$ | $65.5 \pm 1.0$ | $6.3 \pm 0.8$ |
| Krytox 107 | 450 | $120.9 \pm 0.4$ | $107.8 \pm 1.0$ | $13.1 \pm 1.3$ | $73.5 \pm 0.9$ | $61.1 \pm 1.8$ | $12.4 \pm 1.3$ |
| Hexadecane | 2.8 | $106.2 \pm 2.5$ | $103.2 \pm 1.8$ | $3.0 \pm 0.8$ | $\approx 0$ | $\approx 0$ | N/A |

All coatings infused with PFPE displayed excellent amphiphobicity, as both water and n-hexadecane drops did not stick to these surfaces leading to low contact angle hysteresis values. The increase in CAH observed for PFPE-infused SLIPS from Krytox 100 to 107 has already been reported in the literature [28] and is due to the increasing chain length of PFPE molecules, which leads to higher oil viscosity: the recession of water drops on these surfaces is hindered by pinning of their contact line, thus diminishing drop mobility and receding contact angle [29].

Hexadecane-infused SLIPS also showed excellent mobility of water drops, with a $CAH_W$ of 3.0°. Obviously, these surfaces were completely oleophilic as hexadecane drops quickly wetted the surface, therefore it was not possible to measure contact angles with hexadecane.

### 3.2. Durability Tests

3.2.1. Response to UV Irradiation

All fabricated SLIPS were exposed to UV radiation and their wetting behavior was evaluated after 2, 4, 6, and 8 h of irradiation. However, as the changes in contact angles were limited, only the values measured after 8 h are reported for brevity. Figure 2 compares the CAH values of the fabricated SLIPS before and after UV irradiation.

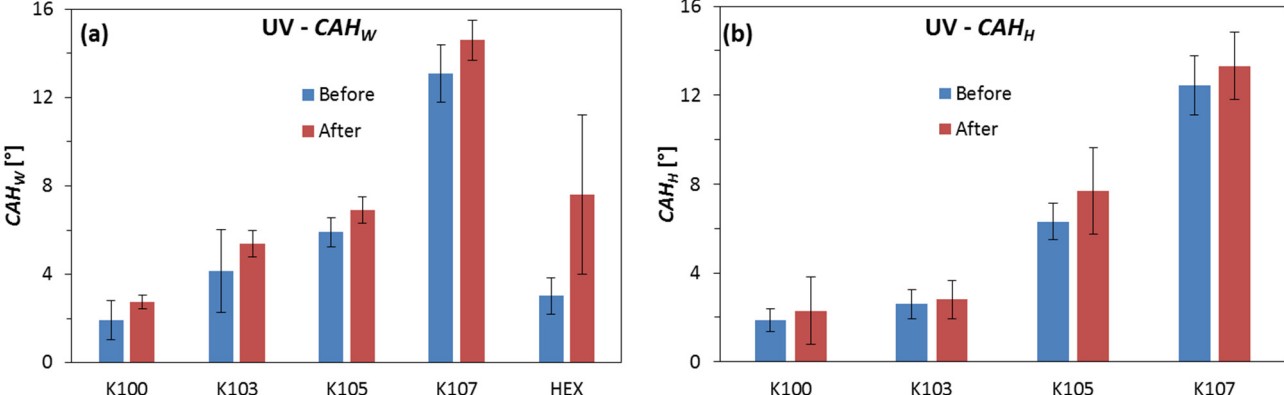

**Figure 2.** Contact angle hysteresis with (**a**) water (CAH$_W$) and (**b**) n-hexadecane drops (CAH$_H$) for SLIPS samples before (blue) and after UV irradiation (red) for 8 h. Standard deviations are reported as error bars.

Krytox-infused SLIPS had almost constant CAH values with both liquids, thus displaying excellent resistance to prolonged UV exposure. These liquids have limited UV absorption [30], therefore they do not undergo radiolysis. Moreover, even though the actual values of vapor pressure for commercial Krytox lubricants are unknown, PFPE usually possesses extremely low vapor pressure, in the order of $10^{-8}$ Torr at 20 °C [31].

On the other hand, hexadecane-infused SLIPS behaved differently. After 6 h, contact angle hysteresis with water (CAH$_W$) increased from 2.6° to 3.8°; then, after 8 h, it further increased to 7.6° along with standard error (from 0.2° to 3.6°). Both phenomena suggest the formation of defects on the SLIPS, probably due to evaporation of the infused oil which has significant vapor pressure at room temperature (1.4 $\times$ $10^{-4}$ Torr) [32]. The underlying surface might have been exposed, acting as a surface defect point that caused the increase in CAH$_W$. These results suggest that hexadecane-infused SLIPS are not ideal candidates for applications in which the surface is exposed to air for prolonged time, due to the high evaporation rate of the lubricant. Notably, hexadecane is the least volatile among alkanes that are liquid at room temperature; therefore, the same considerations apply to the lower homologues of hexadecane (i.e., tetradecane, dodecane, decane, etc.). On the contrary, PFPE-based SLIPS showed remarkable stability when exposed to UV and represent a good choice for application on surfaces that remain often dry.

### 3.2.2. Response to Chemical Ageing

In order to evaluate their resistance to chemically aggressive environments, SLIPS samples were immersed in either acidic, alkaline, or saline solutions and their wetting properties were evaluated after different immersion times. Figure 3 recaps the behavior of SLIPS samples in terms of CAH with water (left-hand graph) and n-hexadecane drops (right-hand graph).

In both cases, K100 samples proved remarkably less durable than other Krytox-infused samples. Despite their low initial CAH$_W$ and CAH$_H$ values, after only 3 days of immersion in the acidic solution their amphiphobicity was lost, with CAH$_W$ = 89° and CAH$_H$ = 41°. After 14 days of immersion, RCA values with both liquids were rapidly decreasing, therefore data acquisition was interrupted. This behavior indicates that the infused Krytox 100 oil was removed by the acidic solution, thus exposing the underlying coating and degrading it as suggested by the low receding contact angle values with water (RCA$_W$ = 39°) and hexadecane (RCA$_H$ = 19°). On the contrary, the other Krytox-infused SLIPS showed significant durability in the acidic environment. Notably, increasing the viscosity of the infused liquid led to improved retention of the amphiphobic behavior, with K107 sample being the most stable SLIPS. It is worth to highlight the different trends of CAH with water and n-hexadecane. The former increases with time, probably due to the formation of polar -OH hydroxyl groups in PFPE chains which increase the interaction with water molecules via hydrogen bonding. To confirm this hypothesis, we performed FTIR

analysis on the fabricated surfaces. First, the spectrum of each surface was obtained prior to immersion in the acidic solution, then it was acquired after 7 and 60 days of immersion. Figure 4 reports the spectra obtained for K107 as an example.

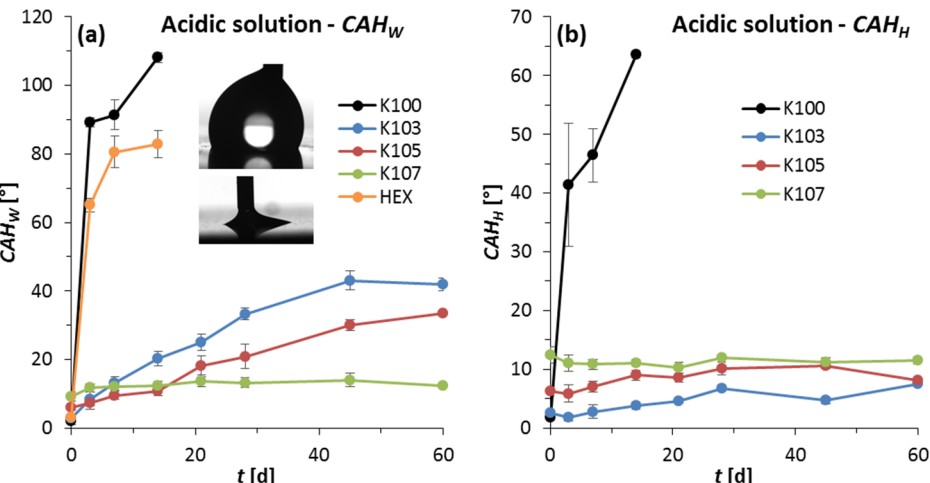

**Figure 3.** Contact angle hysteresis with (**a**) water (CAH$_W$) and (**b**) n-hexadecane drops (CAH$_H$) for SLIPS samples immersed in an acidic solution (pH = 3) as a function of immersion time t: K100 (black), K103 (blue), K105 (red), K107 (green), and HEX (orange). Standard deviations are reported as error bars. Inset: examples of frames for the measurement of RCA$_W$ on K107 (top) and HEX samples (bottom) after 7 days of immersion.

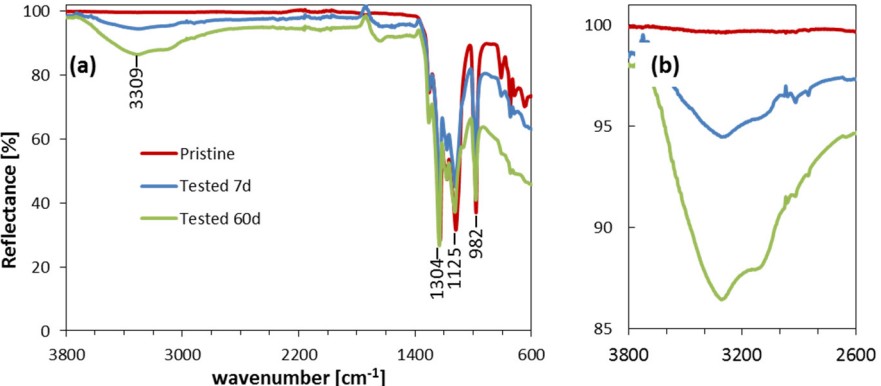

**Figure 4.** (**a**) Fourier Transform Infrared spectra (FTIR) of K107 sample before (red) and after immersion in acidic solution for 7 (blue) and 60 days (green). (**b**) Detail of the spectra in the 3800–2600 cm$^{-1}$ range. The positions of the most relevant peaks are reported.

After 7 days of immersion, a broad band centered at 3309 cm$^{-1}$ appeared; it can be assigned to the stretching vibration of -OH groups formed on polyether chains [33,34]. Notably, after 60 days the intensity of the band increased, as highlighted in the inset spectra, suggesting that further hydroxylation of the chain had occurred. Fourier Transform IR (FTIR) spectra for all Krytox-infused surfaces after 7 days are compared in Figure 5.

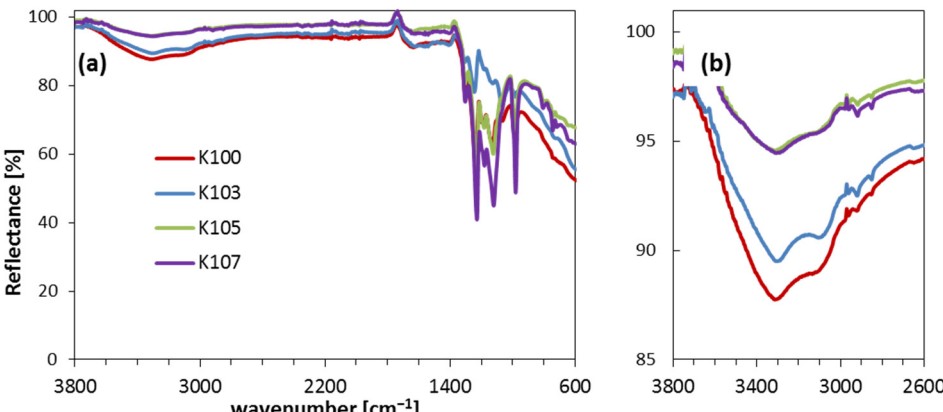

**Figure 5.** (**a**) Fourier Transform Infrared spectra (FTIR) of K100 (red), K103 (blue), K105 (green), and K107 samples (purple) after immersion in acidic solution for 7 days. (**b**) Detail of the spectra in the 3800–2600 cm$^{-1}$ range.

Focusing on the -OH stretching band (Figure 5b), it is clear that K100 suffered the highest degree of hydroxylation, followed by K103, while K105 and K107 showed similar signal. It was not possible to compare FTIR spectra after more prolonged immersion time due to the severe degradation of K100 samples.

On the other hand, $CAH_H$ did not change significantly, probably because the aforementioned polar -OH groups do not interact with non-polar hexadecane molecules, thus they do not affect contact angle values. Moreover, the different size of water and n-hexadecane molecules can contribute to explain this phenomenon. According to Wang et al. [35], small water molecules can penetrate the damaged PFPE network, resulting into pinning phenomena and increased $CAH_W$; on the other hand, large n-hexadecane molecules cannot do the same and $CAH_H$ remains substantially unaltered.

Regarding HEX samples, they showed a quick increase in $CAH_W$, probably due to n-hexadecane displacement and degradation of the underlying coating as observed for K100. In fact, n-hexadecane is even less viscous than Krytox 100 and can be displaced by water shortly after immersion.

Similar CAH trends were observed for SLIPS immersed in alkaline solution, as displayed in Figure 6.

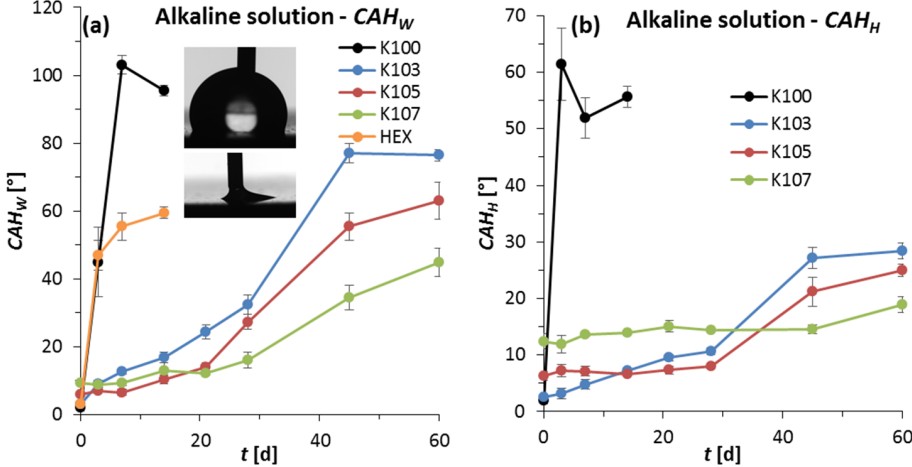

**Figure 6.** Contact angle hysteresis with (**a**) water ($CAH_W$) and (**b**) n-hexadecane drops ($CAH_H$) for SLIPS samples immersed in a alkaline solution (pH = 11) as a function of immersion time t: K100 (black), K103 (blue), K105 (red), K107 (green), and HEX (orange). Standard deviations are reported as error bars. Inset: examples of frames for the measurement of RCA$_W$ on K107 (top) and HEX samples (bottom) after 7 days of immersion.

K100 and HEX samples showed quick degradation of their liquid-repellent behavior, with both CAH values rapidly increasing in few days. The unexpected drop in CAH for K100 after 7–14 days is due to the fact that ACA decreases more than RCA in that period. For other PFPE-infused SLIPS, the increase in $CAH_W$ was more evident than in the acidic environment: $CAH_W$ value for the K103 sample reached 77° after 45 days. Increasing lubricant viscosity led to smaller increase in $CAH_W$, with K107 displaying the best value at 45° after 60 days. As described before for the immersion in acidic solution, $CAH_H$ increased less than $CAH_W$, also, in alkaline conditions, and the same trend with viscosity was observed, as $CAH_H$ for K107 remained constant after 60 days. FTIR spectra were collected after 7 days (Figure 7).

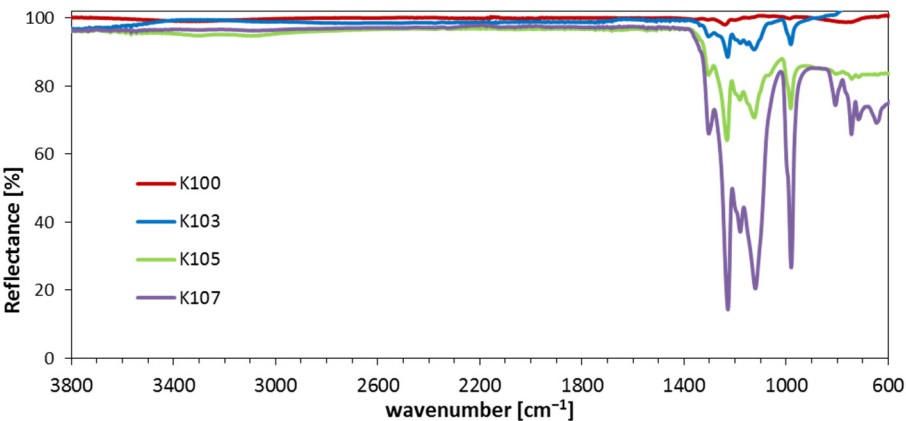

**Figure 7.** Fourier Transform Infrared spectra (FTIR) of K100 (red), K103 (blue), K105 (green), and K107 samples (purple) after immersion in alkaline solution for 7 days.

FTIR spectra confirm that less viscous PFPE oils are more prone to degradation than the more viscous ones: K100 gave no signal, suggesting a complete loss of PFPE; on the other hand, increasing oil viscosity led to more intense signal below 1400 cm$^{-1}$, with K107 being unaltered (see Figure 4 for comparison). From these results, it seems clear that PFPE is more susceptible to alkaline environments than to acidic ones, as already reported in the literature [36]. Indeed, polyether chains are intrinsically hydrophilic and prone to hydrolysis in alkaline conditions [37]; probably, substitution of the polymer backbone with fluorine atoms might only temporarily delay hydrolysis. Judging from FTIR spectra, low-viscosity PFPE oils are totally depleted in alkaline conditions, while in acidic solution they, rather, seem to be hydroxylated but not removed.

The immersion tests in saline solution showed similar trends to those observed for the alkaline conditions (Figure 8).

Once again, K100 and HEX quickly lost their amphiphobicity, with steep increase in CAH after only 3 days; the decrease in $CAH_H$ for K100 was due to the drop in $ACA_W$. Meanwhile, K103 lost its amphiphobicity more gradually, stabilizing its CAH values after 28–45 days; on the other hand, K105 and K107 retained their wetting properties for the entire testing period. Especially K107 had its CAH values almost completely unaltered after 60 days of immersion in the saline solution. FTIR spectra after 7 days (not reported for brevity) were compared also for these samples; as observed in alkaline solution, lubricant depletion increased with decreasing PFPE viscosity. Such result is remarkable in perspective of an application of these coatings in marine environment and in corrosive conditions in general; the retention of the amphiphobic behavior indirectly suggests anti-corrosion properties for K107 coatings. The anti-corrosion properties of SLIPS have already been reported [38,39], but never in such harsh conditions as usually mild NaCl solutions or seawater are used; in our tests, NaCl concentration was almost three times larger than in seawater, therefore corrosion is expected to be more severe.

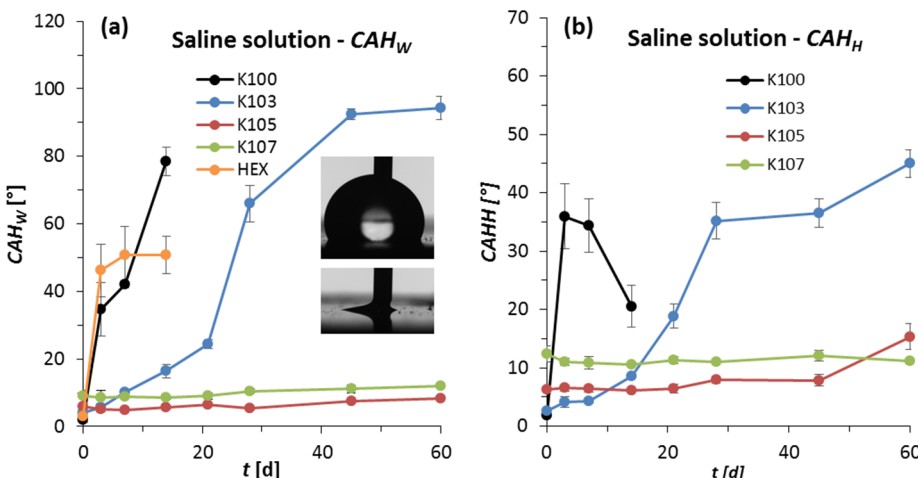

**Figure 8.** Contact angle hysteresis with (**a**) water (CAH$_W$) and (**b**) n-hexadecane drops (CAH$_H$) for SLIPS samples immersed in a saline solution (NaCl 100 g L$^{-1}$) as a function of immersion time t: K100 (black), K103 (blue), K105 (red), K107 (green), and HEX (orange). Standard deviations are reported as error bars. Inset: examples of frames for the measurement of RCA$_W$ on K107 (top) and HEX samples (bottom) after 7 days of immersion.

### 3.2.3. Response to Abrasion

Mechanical stresses are the most common cause of performance loss in liquid-repellent coatings; therefore, it is necessary to address their response to such stresses in perspective of future applications. We chose to perform abrasion tests as per the UNI EN 1096-2 standard because it is widely applied on coatings for the building industry; moreover, this test applies compression and shear stress on the surface contemporarily, thus effectively simulating complex operational conditions. CAH values for the tested SLIPS before and after abrasion tests are reported in Figure 9.

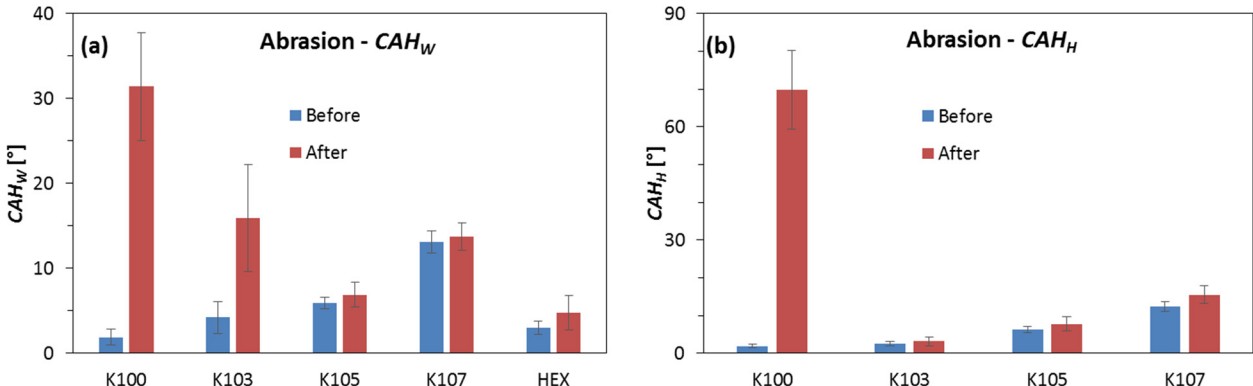

**Figure 9.** Contact angle hysteresis with (**a**) water (CAH$_W$) and (**b**) n-hexadecane drops (CAH$_H$) for SLIPS samples before (blue) and after abrasion (red) as per the UNI EN 1096-2 standard. Standard deviations are reported as error bars.

Among PFPE-infused SLIPS, a relationship between lubricant viscosity and increase in CAH$_W$ was observed (Figure 10): the sample infused with the least viscous oil (K100) showed the most significant increase in CAH values, eventually losing its amphiphobicity.

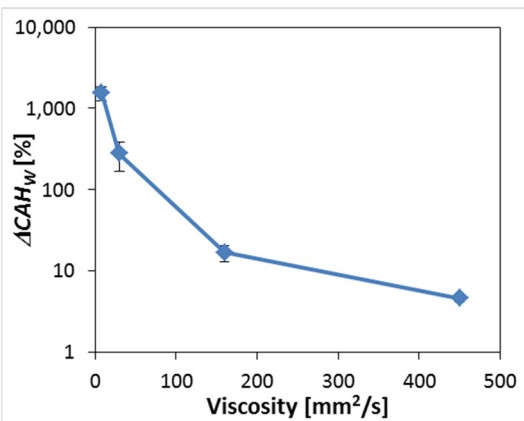

**Figure 10.** Increase in contact angle hysteresis with water ($\Delta CAH_W$) as a function of oil viscosity for PFPE-infused SLIPS. Standard deviations are reported as error bars.

With increasing lubricant viscosity, the increase in CAH (especially with water) became less evident, with K105 and K107 samples retaining their amphiphobicity after the abrasion tests. In the past years, several papers [40,41] investigated the response of PFPE oils under abrasion because of their application in hard disk drives; it was demonstrated that, in such conditions, these materials can be involved in tribochemical degradation reactions, which can be significantly catalyzed by Lewis acids like $Al_2O_3$ [42]. The most important degradation mechanisms include triboelectrical reactions (with creation of radical species) and mechanical cleavage, due to the friction between PTFE and solid surface asperities [43]. The degradation rate depends on the length of polymer chains: short macromolecules like those in K100 have higher mobility (i.e., lower viscosity) which lead to higher degradation rates. Increasing the chain length can slow down degradation reactions (especially mechanical cleavage) [44], thus explaining the retention of amphiphobic properties of K105 and K107. It is also necessary to consider that abrasion tests cause friction and related increase in surface temperature; Krytox 100 is more susceptible to temperature increase than higher Krytox oils, with obvious negative effect on the amphiphobicity of K100.

On the other hand, HEX samples showed limited increase in $CAH_W$ after the test, although n-hexadecane has lower viscosity than Krytox 100. These results can be explained considering that the C-C bonds in the n-hexadecane molecule are less prone to mechanical cleavage than the C-O bonds in PFPE polymer chain [45]; therefore, HEX SLIPS are less prone than K100 ones to mechanical degradation caused by abrasion.

## 4. Conclusions

We reported the fabrication of SLIPS based on a nanostructured, porous alumina coating infused with low surface tension lubricants with different chemical composition (PFPE, n-hexadecane) and viscosity (from 2.8 to 450 mm$^2$/s). Wetting characterization of as-produced SLIPS showed that contact angle hysteresis with water and hexadecane drops increased with the viscosity of the infused liquid. However, higher viscosity led to enhanced resistance to almost all of the tested severe environmental conditions: n-hexadecane and the least viscous PFPE (Krytox 100) proved unstable in acidic, alkaline, and saline environments, leading to complete loss of repellence in few days. On the other hand, SLIPS infused with the most viscous PFPE oil (Krytox 107) retained their amphiphobic behavior for up to 60 days, especially in acidic and highly saline solutions. These results lead to consider these surfaces as potential candidates for application in the naval and maritime industries. The same trend was observed in abrasion tests: the high mobility of short polymer chains in the low-viscosity Krytox 100 made them susceptible to tribochemical degradation, which in turn did not affect the repellence of highly viscous Krytox 107. Even the chemical composition of the infused liquid affected SLIPS durability under abrasion: despite its very low viscosity, hexadecane proved stable in such conditions

due to the non-cleavable C-C bonds forming its chain. These considerations are extremely relevant for the application of SLIPS in devices like hard drives or lubricated joints for mechanics. On the other hand, hexadecane-infused SLIPS proved the least stable in UV irradiation tests due to the high volatility of the infused lubricant, which makes these coatings unsuitable for applications in which the surface is expected to be exposed to air (i.e., not being wetted by another liquid or in an open ambient). These results once again highlight the need for careful consideration of the application conditions when designing a liquid-infused coating.

**Author Contributions:** Conceptualization, F.V. and M.R.; methodology, F.V. and M.R.; software, F.V.; validation, F.V. and A.C.; formal analysis, F.V.; investigation, G.G.; resources, F.V. and M.R.; data curation, F.V. and G.G.; writing—original draft preparation, F.V.; writing—review and editing, F.V., A.C., and M.R.; visualization, F.V. and A.C.; supervision, F.V. and M.R.; project administration, F.V. and M.R.; funding acquisition, M.R. All authors have read and agreed to the published version of the manuscript.

**Funding:** This research received no external funding.

**Institutional Review Board Statement:** Not applicable.

**Informed Consent Statement:** Not applicable.

**Data Availability Statement:** Data available in a publicly accessible repository. The data presented in this study are openly available in https://doi.org/10.6084/m9.figshare.16566960.v1, accessed on 6 September 2021.

**Acknowledgments:** The authors want to acknowledge the entire Smart Surfaces Group at CNR-ISTEC for their precious work.

**Conflicts of Interest:** The authors declare no conflict of interest.

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
