# Peer review of "Evaluation of the Durability of Slippery, Liquid-Infused Porous Surfaces in Different Aggressive Environments: Influence of the Chemical-Physical Properties of Lubricants"

_coatings, doi:10.3390/coatings11101170_

Round 1

Reviewer 1 Report

Federico Veronesi, et al. investigated the durability of slippery, liquid-infused porous surfaces in different aggressive environments. I have some suggestions for a minor revision.

For Fig. 4, Fig. 5, and Fig. 7, please mark the wavelength of the main peaks in the figures.

If possible, please show the photos of representative measurement of contact angles in Fig. 3, Fig. 6, and Fig. 8.

Please provide error bars in Fig. 10.

Author Response

We thank the Reviewer for the precious comments.

  • We added the position of the peaks in Figure 4. Moreover we added the sentence "The positions of the most relevant peaks are reported" to the caption. Peak positions did not change in the following spectra (i.e., in Figures 5 and 7), thus they were not reported.
  • We added examples of frames for RCAW measurement on K107 and HEX samples after 7 days of immersion in each aggressive solution, with the aim of showing how the dewetting properties of HEX deteriorate while K107 is still dynamically hydrophobic. In the captions of Figures 3, 6 and 8 we added the phrase "Inset: examples of frames for the measurement of RCAW on K107 (top) and HEX samples (bottom) after 7 days of immersion."
  • We added error bars to Figure 10, alongside with the phrase "Standard deviations are reported as error bars" in the caption.

Reviewer 2 Report

The article looks at the durability of SLIPS in various aggressive environments. The sample preparation and investigations are generally well described, but unfortunately no detailed information on the production of the nanostructured pseudo-boehmite is given. Here only a reference is given and the interested reader must search for the literature mentioned in this article.

The presented results are easy to understand and support the discussion. The results are of practical importance.

The integration of the figures and tables did not work in the entire article (see lines 155, 159 etc. “Error! Reference source not found…”). That needs to be corrected.

Results and Discussion

Lines 164 and 359:
Please use the SI system instead of the cgs system for the unit of measurement for the kinematic viscosity.

Line 186:

The CAH values differ slightly. The phrase “Krytox-infused SLIPS had almost constant CAH values …” is more appropriate.

Line 234:

FTIR-ATR spectra are shown in Figure 5, where the transmittance is plotted versus the wavenumber. Correct the ordinate labeling and add the scaling of the ordinate. For a better comparison, the baseline-corrected spectra should be normalized (baseline 100% transmittance) e. g. to a C-F stretching vibration band.

Line 280:

Correct the ordinate labeling and add the scaling of the ordinate in Figure 7.

  Author Response

Thanks to the Reviewer for the helpful suggestions.

  • Concerning the , We chose not to report the detailed fabrication process of the nanostructured pseudo-boehmite layer as the authors already described it in several past papers. In general, the reaction between aluminum/aluminum oxide and hot water to form pseudo-bohemite with a flower-like morphology has been extensively investigated by several authors and was not the focus of the paper, therefore we chose to omit it;
  • We apologize for the inconvenience, references look ok in the .doc but they were not in the .pdf. We fixed all links to Figures and Tables throughout the manuscript by manually rewriting them;
  • We changed "cSt" to "mm2/s" in Table 1 and in Conclusions;
  • We added "almost" at line 186.
  • Figures 4, 5, and 7 report Reflectance on the ordinate, as all spectra were collected in ATR mode on coated aluminum samples. Ordinate scales from 0 to 100% were added in the aforementioned Figures. Please note that, for each spectrum, the non-infused face of the sample (i.e. the rear) was used as background, therefore the recorded spectra are only originated by the infused oil and potential degradation products. FTIR spectra are supposed to give limited information on the degradation/depletion of the infused oil and are not meant to be quantitative.

Reviewer 3 Report

The manuscript “Evaluation of the Durability of Slippery, Liquid-Infused Porous Surfaces in Different Aggressive Environments: Influence of the Chemical-Physical Properties of Lubricants” has been carefully reviewed. The research results are interesting, and the manuscript is well organized and written in good English. Nevertheless, there are a lot of references not correctly citied in the manuscript that showed “Error! Reference source not found” in the manuscript located at Page 4 (Line 155 and159), Page 5 (Line 180 and 208), Page 6 (Line 232), Page 7 (Line 242 and 264), Page 8 (Line 279 and 286), Page 9 (Line 295 and 323) and Page 10 (Line 329). In addition, the 35th reference (located at Page 36, Line 474) is shown in an incorrect citation format. It is recommended that the authors carefully revise and proofread the manuscript before resubmitting it.

Author Response

Thanks to the Reviewer for the helpful comments.

  • We sincerely apologize for the format errors, the manuscript looks ok to us ad .doc  but it does not as .pdf. To fix this issue, we manually rewrote all references to Figures and Tables throughout the manuscript.
  • Concerning Ref.35, we checked it and the format seems ok to us. We rather found a couple errors in Ref.36 and fixed them. Here are Ref.34, 35 and 36, please let us know if you find any other error.

34. Zheng, S.; Al, S.; Guo, Q. Poly(Hydroxyether of Phenolphthalein) and Its Blends with Poly(Ethylene Oxide). Polym. Sci. Part B Polym. Phys. 2003, 41, 466–475, doi:10.1002/polb.10392.

35. Wang, Y.; Dugan, M.; Urbaniak, B.; Li, L. Fabricating Nanometer-Thick Simultaneously Oleophobic/ Hydrophilic Polymer Coatings via a Photochemical Approach. Langmuir 2016, 32, 6723–6729, doi:10.1021/acs.langmuir.6b00802.

36. Yang, Y.-W.; Hentschel, J.; Chen, Y.-C.; Lazari, M.; Zeng, H.; Michael Van Dam, R.; Guan, Z. “‘Clicked’” Fluoropolymer Elastomers as Robust Materials for Potential Microfluidic Device Applications. Mater. Chem. 2012, 22, 1100, doi:10.1039/c1jm14131g.